# A Clinical Case of a Homozygous Deletion in the *APOA5* Gene with Severe Hypertriglyceridemia

**DOI:** 10.3390/genes13061062

**Published:** 2022-06-14

**Authors:** Petr Andreevich Vasiluev, Olga N. Ivanova, Natalia A. Semenova, Tatiana V. Strokova, Natalia N. Taran, Uliana V. Chubykina, Marat V. Ezhov, Ekaterina Y. Zakharova, Elena L. Dadli, Sergey I. Kutsev

**Affiliations:** 1Research Centre for Medical Genetics, Moskvorechye Street, 1, 115522 Moscow, Russia; ion10@bk.ru (O.N.I.); semenova@med-gen.ru (N.A.S.); doctor.zakharova@gmail.com (E.Y.Z.); dadali@med-gen.ru (E.L.D.); kutsev@mail.ru (S.I.K.); 2Institute of Gene Biology, The Russian Academy of Sciences, 34/5 Vavilova Street, 119334 Moscow, Russia; 3Federal Research Centre of Nutrition and Biotechnology, Kashirskoe Shosse, d. 21, 115446 Moscow, Russia; strokova_t.v@mail.ru (T.V.S.); pknt@mail.ru (N.N.T.); 4Federal State Budget Organization «Chazov National Medical Research Center of Cardiology» of the Ministry of Health of the Russian Federation, 3rd Cherepkovskaya Street 15a, 121552 Moscow, Russia; uliankachubykina@gmail.com (U.V.C.); marat_ezhov@mail.ru (M.V.E.)

**Keywords:** hypertriglyceridemia, pancreatitis, cardiovascular disease, *APOA5*, *LPL*, mutation, gene panel

## Abstract

Background: Hypertriglyceridemia (HTG) is one of the most common forms of lipid metabolism disorders. The leading clinical manifestations are pancreatitis, atherosclerotic vascular lesions, and the formation of eruptive xanthomas. The most severe type of HTG is primary (or hereditary) hypertriglyceridemia, linked to pathogenic genetic variants in *LPL*, *APOC2*, *LMF1,* and *APOA5* genes. Case: We present a clinical case of severe primary hypertriglyceridemia (TG level > 55 mmol/L in a 4-year-old boy) in a consanguineous family. The disease developed due to a previously undescribed homozygous deletion in the *APOA5* gene (NM_052968: c.579_592delATACGCCGAGAGCC p.Tyr194Gly*68). We also evaluate the clinical significance of a genetic variant in the *LPL* gene (NM_000237.2: c.106G>A (rs1801177) p.Asp36Asn), which was previously described as a polymorphism. In one family, we also present a different clinical significance even in heterozygous carriers: from hypertriglyceridemia to normotriglyceridemia. We provide evidence that this heterogeneity has developed due to polymorphism in the *LPL* gene, which plays the role of an additional trigger. Conclusions: The homozygous deletion of the *APOA5* gene is responsible for the severe hypertriglyceridemia, and another SNP in the *LPL* gene worsens the course of the disease.

## 1. Introduction

Hypertriglyceridemia (HTG) is a lipid metabolism disorder occurring when the level of triglycerides (TG) in the blood plasma exceeds 1.7 mmol/L (150 mg/dL). Average TG levels range from 1.45 (128 mg/dL) to 1.25 mmol/L (110 mg/dL) for adult men and women, respectively [1]. In children, the concentration of TG is normally lower than in adults, and there is a heterogeneity of TG levels among subjects of different ethnicities [2,3]. According to data from the European Societies of Cardiology and Atherosclerosis Guidelines, mild and moderate HTG are limited by a TG level from 1.7 to 10 mmol/L, while severe HTG occurs at ≥10 mmol/L [4]. As of 2020, HTG is considered the most common type of dyslipidemia, occurring in 16% of the total American population [5]. Additionally, among Mexicans, 31% of people have mild to moderate HTG, and about 5% of the population have severe hypertriglyceridemia, with plasma levels over 10 mmol/L [6]. According to the Russian study “ESSE-RF” (The Epidemiology of Cardiovascular Risk Factors and Diseases in Regions of the Russian Federation), which assessed the distribution of lipid levels in men and women of working age in 13 regions of Russia, a strong increase in TG levels (>5.0 mmol/L) was found in 1.1% of the population [7].

According to the Framingham Study, and other publications, a TG level over 1.7 mmol/L is associated with a significantly higher risk of cardiovascular events [8,9,10,11]. In patients with a concentration of TG more than 2.3 mmol/L and, at the same time, a high-density lipoprotein cholesterol (HDL-C) level of less than 0.8 mmol/L, the risk of cardiovascular events increases 10-fold compared with patients with normal TG and HDL-C levels [12].

The main mechanism underlying atherosclerotic lesions during HTG is the overproduction of very-low-density lipoproteins (VLDL) in the liver. In atherogenic combined dyslipidemia, it has been noted that there is a transfer of TG from VLDL to low-density lipoproteins (LDL), and at the same time, a transfer of cholesterol esters from LDL to VLDL. This phenomenon leads to the appearance of an additional number of atherogenic lipoproteins: VLDL have lost some TG and become smaller, thus increasing atherogenicity [13].

In addition, the leading clinical manifestation of familial HTG is acute pancreatitis [14,15,16], which can develop even in a patient with a TG level of 5–10 mmol/L (440–880 mg/dL) [17]. In this case, the higher the TG level in the blood, the higher the risk of pancreatitis. The mechanism of pancreatitis most likely consists in chylomicronemia and blood hyper-viscosity, reducing the capillary blood flow in the pancreas and thus causing ischemia, acidification, damage of pancreatic cells, and activation of inflammatory processes.

The etiology of HTG varies. Primary or hereditary HTG are genetically determined and marked by clinical variability and heterogeneity. Appendix A provides a list of the most common types of hereditary HTG and the genes responsible for their development. In addition, a number of diseases and conditions can lead to a secondary increase of TG levels in blood plasma. These include obesity, diabetes mellitus, metabolic syndrome, hypothyroidism, Cushing syndrome, nephrotic syndrome, diet patterns, alcohol abuse, smoking, and uptake of certain medications [18,19,20,21,22,23,24].

## 2. Clinical Case

The clinical case studies a male child, four years old, Avar by nationality, born from a third pregnancy in the family from a consanguineous marriage: father and mother are fourth cousins (Figure 1). He has healthy older brothers of 11 and 6 years old. After birth, his weight and length were 4000 g and 55 cm, with an Apgar score of 8/8. There is a history of dyslipidemia in the family: one sibling, father, aunt (paternal), and uncle (paternal) have increased blood TG levels (Table 1). They did not take any lipid-lowering medications and did not have any cardiovascular disease.

During a routine examination of the proband at the age of 1.5 years, serum lactescence was revealed with a TG level up to 55 mmol/L (4850 mg/dL). Later, while dyslipidemia remained, an increase in liver enzymes was found (ALT up to 61 U/L, AST up to 49 U/L). Ultrasound examination revealed hepatomegaly, splenomegaly, and pancreatic inflammation. During several hospitalizations, serum lactescence and HTG have been reported and have not been eliminated by a low-fat diet (Table 2). In addition, the patient received therapy with iron, ursodeoxycholic acid, antihistamines, interferon, and intestinal microflora stabilizers.

The patient was referred to the Moscow Research Center for Medical Genetics, named after the academician N.P. Bochkov, to clarify the diagnosis. On examination, the patient did not have any stigmas of dysembryogenesis, but on the skin of his hands, there was a light rash, and on his legs, there was a reddish rash of the xanthoma type, in total more than 10 elements. On the skin of the dorsum of his hands and in the area of the first finger, whitish elements of a rounded shape (draining in places) up to 5 mm in diameter, resembling xanthomas, were noted. In the sacral region and on the inner surface of the legs, there were single reddish elements, also resembling xanthomas (Figure 2).

Initially, an analysis of the *LPL* gene was carried out because genetic variants in this gene are associated with impaired metabolism of TG. We found an exonic variant: *LPL* (NM_000237.2: c.106G > A (rs1801177) p.Asp36Asn), in a heterozygous state. This variant is described in HGMD (CM910259) and ClinVar as associated with an increased risk of cardiovascular disease and is noted in the UniProtKB database as associated with familial type 3 hyperlipidemia (P06858). The frequency of this variant according to GnomAD data is 0.014733.

This genetic variant was initially regarded as a polymorphism, which should not have led to clinical manifestations of HTG. To verify the genetic substrate, an additional analysis of 60 genes responsible for the development of hereditary dyslipidemias was carried out (see Appendix A for the list of genes included in our custom panel). According to the analysis, the presence of a homozygous deletion of 13 nucleotides in size in exon 4 of the *APOA5* gene (NM_052968: c.579_592delATACGCCGAGAGCC) was suggested, leading to the formation of a premature stop codon, p.Tyr194Gly*68.

Subsequently, the deletion was confirmed by direct sequencing in a homozygous state in the proband and in a heterozygous state in both parents (Figure 3A–C, respectively). This deletion was also found in the 11-year-old brother, while the second sibling and other relatives were unavailable for examinations (Figure 3D).

Thus, based on the clinical picture and the results of biochemical and genetic analyses, familial hyperchylomicronemia was diagnosed in this patient (ICD10: E 78.3, OMIM: 144650, Fredrickson classification: type 1) due to a previously undescribed homozygous deletion in the *APOA5* gene.

## 3. Materials and Methods

Ion S5 (Thermo Fisher Scientific, Inc., Waltham, MA, USA) sequencing systems were used. Genomic DNA (gDNA) was extracted from whole blood with the use of a Diatom DNA Prep reagent kit (Biocom, San Diego, CA, USA) according to the manufacturer’s protocols. The concentration of gDNA as well as DNA concentrations of the libraries afterwards were determined using a Qubit™ Fluorometer (Thermo Fisher Scientific, Inc.).

DNA libraries were constructed with the Ion AmpliSeq™ targeted custom panel and the Ion AmpliSeq™ Library kit 2.0 (Thermo Fisher Scientific, Inc., cat. Nos. 04779971_Dyslipidemia_IAD175748_182 and 4480442, respectively). The panel contains exons of 60 genes, including genes for primary monogenic hypertriglyceridemia and hypercholesterolemia (see Appendix A).

DNA quality was confirmed by the final stage of library preparation using the PCR test, performed with the manufacturer’s primers to adaptors’ sequences. The thermocycling conditions used were as follows: 95 °C for 40 s, 68 °C for 35 s, and 72 °C for 75 s, and the number of cycles used was dependent on the library concentration. PCR results were visualized on silver-stained 8% acrylamide gel (staining time: 10 min at 4 °C). Massive parallel sequencing of pooled libraries with a loading concentration of 75 pmol was performed using an Ion 540™ Chip kit (Thermo Fisher Scientific, Inc., cat. No. A27766) and Ion 540™ Kit-Chef (Thermo Fisher Scientific, Inc., cat. No. A30011), with an average amplicon length of 175 bps. Variant validation was performed by PCR direct sequencing with the Applied Biosystems 3500xL Genetic Analyzer.

Targeted DNA sequencing of the proband supposed the presence of a homozygous deletion in exon 4 of the *APOA5* gene (NM_052968). There were signs of a homozygous deletion of no more than 40 nucleotides (approximate coordinates: c.450_590del). Direct sequencing analysis revealed a homozygous frameshift deletion, c.579_592delATACGCCGAGAGCC, resulting in the premature stop codon p.Tyr194Gly*68.

Bioinformatics Genome Reference Consortium Human Build 37 (GRCh37/hg19) was used for data analysis. The home program for assessing the quality of NGS and variant annotation was used.

The pathogenicity of the variants was assessed in accordance with the recommendations of ACMG Guidelines, 2015 [25]. According to ACMG criteria, the variant is pathogenic (PM2, PVS1, PP3, PP1-M, PP4). The variant is included in ClinVar (by us) (National Center for Biotechnology Information, ClinVar [VCV000995944.1], https://www.ncbi.nlm.nih.gov/clinvar/variation/VCV000995944.1 (accessed on 27 May 2022), rs1940989106). Location: chr11: 116661353–116661366 (GRCh37) UCSC NC_000011.10:116790636:GGCTCTCGGCGTATGG:GG; NG_015894.2:g.6771_6784del; NM_052968.5: c.579_592delATACGCCGAGAGCC; NP_443200.2:p.(p.Tyr194Gly*68).

## 4. Results and Discussion

The *APOA5* gene, located on chromosome 11q23, encodes one of the proteins that builds up lipoproteins and is expressed mainly in the liver and small intestine [26]. Apolipoprotein A5 is included in VLDL cholesterol, HDL cholesterol, and chylomicron particles, and its concentration in blood plasma is extremely low: 20–500 ng/mL [27]. However, this protein is one of the most important modulators of TG metabolism [28], activating lipoprotein lipase [29,30]. It is assumed that apoA5 inhibits the assembly and secretion of VLDL cholesterol in the liver [31] and accelerates the uptake of remnant lipoproteins by the liver [32]. Thus, apoA5 is characterized as a protein that reduces the concentration of TG in blood plasma [29,31,32,33].

The *APOA5* gene consists of 4 exons and 3 introns, with a total protein length of 366 amino acids. The signal sequence is located at the beginning of the protein, at amino acid positions 1 to 23, followed by a long polypeptide chain from amino acid position 24 to 366. Intracellularly, the protein is predominantly located in endosomes and the Golgi apparatus.

ApoA5 is a hydrophobic protein with well-defined functional domains. The extreme C-terminus (mature sequence positions 295–343) facilitates lipid binding, while the N-terminal domain (amino acids 1 to 146 of mature apoA5) adopts a water-soluble helix bundle conformation. The intervening central segment (residues 147–294) possesses a positively charged region (residues 186–227) that is functionally important for lipolysis. The truncated pathogenic variants of the *APOA5* gene are described in HGMD, with the stop codons at positions from 95 to 313 (CM151055).

HTG, caused by pathogenic variants in the *APOA5* gene, is described in OMIM (145750) and HGMD as an autosomal dominant disorder. At the same time, according to ClinVar, only 18 pathogenic/probably pathogenic variants are described in this gene, while according to HGMD, there are 82 variants, and most of them are associated with an increased risk of myocardial infarction and cardiovascular diseases [34,35,36].

According to the existing published research, clinical manifestations in patients with heterozygous pathogenic variants in the *APOA5* gene range from severe hyperchylomicronemia to normotriglyceridemia [35,37]. In some cases, the degree of manifestation is due to the presence of additional triggers, such as obesity, diabetes, or the presence of an additional genetic variant that has a negative effect on TG metabolism. Homozygous pathogenic variants are described relatively rarely and, in most cases, have an extremely severe clinical picture. Thus, Dussaillant et al. described a familial case of a homozygous p.Gln97Ter variant in the *APOA5* gene in two sisters born from a consanguineous marriage in Chile, whose blood TG level was up to 112 mmol/L [35]. In this study, heterozygous carriers of the identified genetic variant had different blood TG levels, from 1.8 to 12 mmol/L. The authors showed that such a difference in the rates of TG metabolism is due to obesity and diabetes in some patients.

The previously undescribed deletion of 13 nucleotides (c.579_592delATACGCCGAGAGCC) in the *APOA5* gene that we discovered leads to the formation of a premature stop codon in the region of amino acid 262 and is localized in the fourth and last exon of this gene. Based on this, the mutant protein most likely avoids NMD (nonsense-mediated mRNA decay) and may be present in the cell. The *APOA5* variant described here is completely lacking the C-terminal lipid-binding motif. This loss of function results in reduced lipolysis and remnant clearance, with the subsequent manifestation of HTG.

In our proband, a homozygous deletion in the *APOA5* gene was inherited from heterozygous parents in a consanguineous marriage, which led to a severe case of homozygous HTG with a 32-fold increase in TG serum concentration. The 5-year-old boy did not develop classical clinical manifestations of severe HTG, such as atherosclerotic vascular lesions, fatty hepatosis, and pancreatitis. However, diffuse changes in the pancreas revealed by ultrasound examination, as well as an increase in hepatic transaminase level, can be regarded as an early onset of damage of the pancreas and liver.

At the same time, the mother of the proband, a carrier of the deletion, has no clinical picture or changes in the lipid profile, while the father of the proband has severe HTG (6.8 mmol/L), and mild HTG (2 mmol/L) is present in the 11-year-old sibling, who is also a carrier of a heterozygous deletion. The diet of different family members does not differ, and the proband’s parents do not have endocrinology or other disorders. No secondary factors that could have led to such different clinical pictures in heterozygous carriers of the mutation have been identified, and therefore we have put forward the concept of the additional influence of the genetic environment. For the parents and siblings, we searched for the variant found in the proband in the *LPL* gene, which was initially considered as an insignificant finding. This variant was identified in the father of the proband and was absent in his mother and brother. Considering that the main effect of the ApoA5 protein is its activating effect on lipoprotein lipase, we assume that even a slight polymorphism in the *LPL* gene, which reduces the activity of this enzyme, in the presence of a pathogenic variant in the *APOA5* gene, will significantly worsen the course of HTG. In the family, the polymorphism in the *LPL* gene seems to be the additional trigger that is necessary for the development of HTG in heterozygous patients. For the proband, this polymorphism has an additional effect, worsening the course of HTG.

## 5. Conclusions

Using our unique diagnostic panel for 60 genes, we have discovered a new pathogenic variant in the *APOA5* gene ((NM_052968: c.579_592delATACGCCGAGAGCC p.Tyr194Gly*68) which, in a homozygous form, leads to severe HTG and early onset of clinical symptoms. In the presence of this variant in a heterozygous state, clinical manifestations vary from average HTG to normotriglyceridemia, which corresponds with published studies. Additionally, we have shown that the presence of a relatively frequent polymorphism in the *LPL* gene worsens the clinical course of primary HTG caused by a mutation in the *APOA5* gene. These results convincingly show that genetic diagnosis is indicated for all patients with suspected hereditary HTG and should include all major genes responsible for lipid metabolism disorders. This approach will make it possible to correctly assess the patients’ risks and prescribe adequate therapy. In addition, we have clearly demonstrated incomplete penetrance in autosomal dominant hypertriglyceridemia caused by a pathogenic variant in the *APOA5* gene.

## Figures and Tables

**Figure 1 genes-13-01062-f001:**
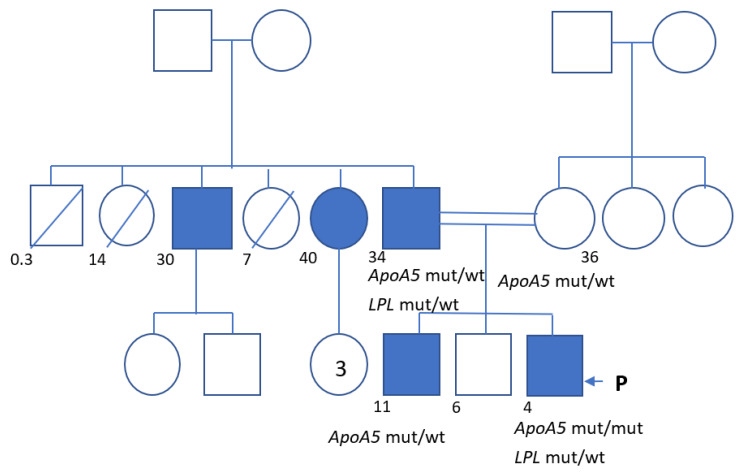
Pedigree of the family we are describing. P—proband. The numbers indicate the age at the time of the initial examination or the time of death.

**Figure 2 genes-13-01062-f002:**
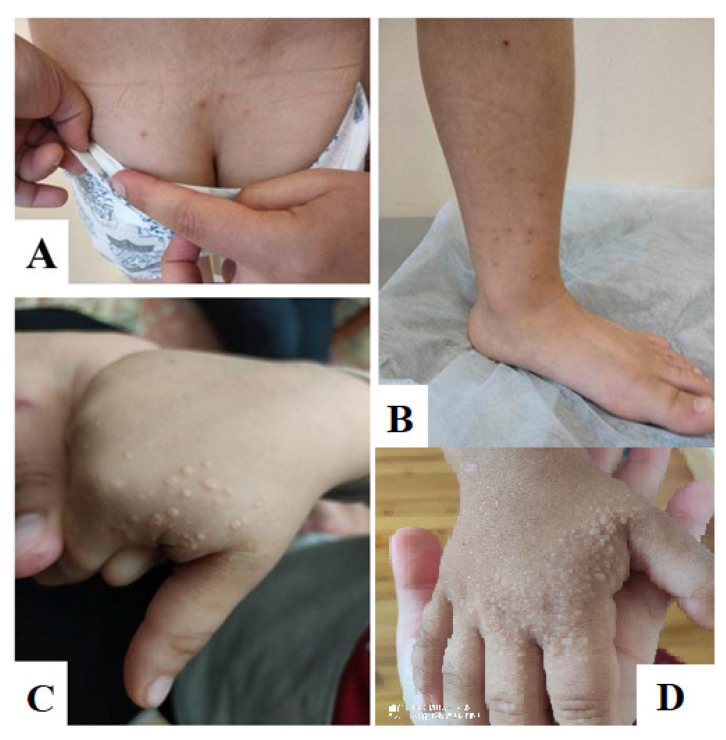
Phenotypic signs of HTG in the patient. (**A**,**B**) Small-cell reddish rash in the lumbar region and lower third of the legs. (**C**,**D**) Small-cell rash on the dorsum of the hands and in the interdigital spaces (**C**) at 4 years old and (**D**) 5 years old.

**Figure 3 genes-13-01062-f003:**
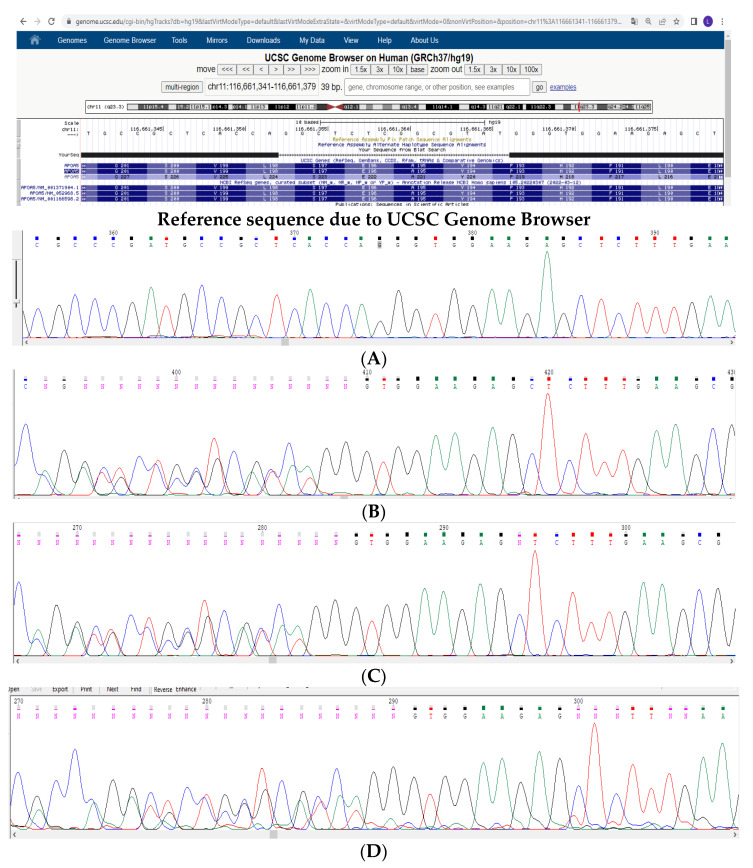
Reference sequence due to UCSC Genome Browser and Sanger sequencing electrograms of: (**A**) proband, (**B**) mother, (**C**) father, and (**D**) sibling (brother).

**Table 1 genes-13-01062-t001:** Lipid levels in patient’s relatives.

	Total Cholesterol	LDL Cholesterol	HDL Cholesterol	Triglycerides	Notes
Normal range	3.2–5.2	1.6–3.8	1.1–2.3	0.1–1.7	
Mother, 36 years old	4.1	2.4	1.3	0.8	—
Father, 34 years old	4.0	1.0	0.7	6.8	Plasma lactescence
Brother, 11 years old	—	—	—	2.0	—
Uncle (p), 30 years old	4.0	1.5	0.5	5.3	—
Aunt (p), 40 years old	5.9	2.0	0.9	7.7	—

Notes: (p)—paternal. Values are in mmol/L. Normal values are based on the most common reference values.

**Table 2 genes-13-01062-t002:** Dynamics of the patient’s lipid levels during diet and medication uptake.

Date	Total Cholesterol	LDL Cholesterol	HDL Cholesterol	Triglycerides	Notes
Normal range	3.2–5.2	1.6–3.8	1.1–2.3	0.1–1.7	
13 January 2017	11.2	—	0.5	55.1	Plasma lactescence
7 February 2017	9.4	—	—	16.1	Plasma lactescence
17 August 2017	3.9	1.2	0.5	6.3	Plasma lactescence

Note: values are in mmol/L.

## Data Availability

Any additional information can be provided by the correspondence author upon request.

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
