# Peer review of "A Clinical Case of a Homozygous Deletion in the APOA5 Gene with Severe Hypertriglyceridemia"

_genes, 2022, doi:10.3390/genes13061062_

Round 1

Reviewer 1 Report

The paper talks about a unique and previously unknown deletion in the APOA5 gene, which increases the clinical pre-disposition of the patient for HTG. However, I think there are some issues with this publication -

  1. It would be great to have a prodigy chart as a part of the main data figure.
  2. The alignment and analysis pipeline of the sequencing data is missing from the article. This doesn't look good as it is the basis of the article.
  3. Can the authors perform in modelling to hypothesize the impact of the amino acid deletions in the APOA5 protein?
  4. Is the panel of genes used in this study alreay available for commercial use?
  5. The method section should have the information about the sequences and not the supplementary materail.

Author Response

Dear reviewer, thank you for your valuable comments on our work. I bring to your attention a new version of the article and the supplement, taking into account the comments of the reviewers.

  1. As I understand you, you recommend inserting the patient's pedigree into the main article, so I put it under the number Fig No. 2.
  2. I took into account your recommendations, so I added a separate section of materials and methods describing the molecular genetic and bioinformatic parts of the work.
  3. In the Results and Discussion section, I added our hypothesis on how gene deletions can lead to HTG. There it is divided into 2 paragraphs, but in general it looks like this. "ApoA5 is a hydrophobic protein with well-defined functional domains. The extreme C-terminus (mature sequence positions 295 – 343) facilitates lipid binding while the N-terminal domain (amino acids 1 to 146 of mature apoA5) adopts a water-soluble helix bundle conformation. The intervening central segment (residues 147 – 294) possesses a positively charged region (residues 186 – 227) that is functionally important for lipolysis. The truncated pathogenic variants of APOA5 are described in HGMD, with the stop codons at positions from 95 to 313 (CM151055). The APOA5 variant described here is completely lack of the C-terminal lipid-binding motif. This loss of function results in reduced lipolysis and remnant clearance, with the subsequent manifestation of HTG"
  4. Yes, this panel can be used commercially, but for this family it was used on a non-commercial basis.
  5. As I wrote above, your recommendations have been taken into account and I have added a separate section of materials and methods to the article.

Thanks again for your comments,

Vasiluev P.

Reviewer 2 Report

Paper for review

Vasiliev et al. report on a clinical case of a homozygous deletion in the APOA5 gene 2 with severe hypertriglyceridemia.

The clinical and molecular data very are very interesting However, there are some Minor concerns that should be added and addressed.

General comments:

  1. The methodology is incomplete. Write techniques that identified a mutation in this gene, gene panel, WES, Sanger, how do you evaluate the pathogenicity? Etc
  2. Make a table about the genes screened in the gene panel and put them in a supplementary file. Put methodology in the main MS.
  3. There is only 1 figure in the MS, why it is labeled as figure 2?
  4. Provide pedigree, Sanger sequencing electro-grams in figure 1.
  5. Gene name should be italic. Kindly correct.
  6. Use three-letter nomenclature for amino acid names.
  7. The authors use the term "mutation(s)". HGVS recommends the use of the neutral term "variant(s)"
  8. The reference sequence used should be described properly, the version number (e.g. NG_0123456. 3; NP_5462.2) is missing. ** Please note that since the reference sequence given (NM_) does not contain intronic sequences a genomic reference sequence should be given as well (e.g. NG_0123456.3).
  9. Gene name should always be italic.
  10. make a flowchart to show the methodology used, so that it becomes easy for the readers to understand and put small headings.

Author Response

Dear reviewer, thank you for your valuable comments on our work. I bring to your attention a new version of the article and the supplement, taking into account the comments of the reviewers.

  1. I took into account your recommendations, so I added a separate section of materials and methods describing the molecular genetic and bioinformatic parts of the work.
  2. A list of genes in tabular form with NM_ is available in the supplement.
  3. I added the patient's pedigree and Sanger sequencing data of the proband and his relatives (Fig. 2 and 3, respectively)
  4. I used the single letter amino acid nomenclature since that was the title of the article I was referring to. I took into account your remark
  5. I used the word "mutation" as a synonym for the "pathogenic genetic variant", but I tried to remove it. Except when I refer to material in another article (Table 1 in supplement)
  6. Since the article is a case report, it is not so important to present the scheme of the experiment here. As for diagnostic methods, if you do not take into account validation, all the main information was obtained by sequencing of gene panel. Thus, in my humble opinion, a graphic image will not bring anything important.

Thanks for your comments, I also corrected the stylistic blots.

best regards,

Vasiluev P.

Round 2

Reviewer 1 Report

The comments were taken care of.